# Risk of Malignancy in Patients with Asthma-COPD Overlap Compared to Patients with COPD without Asthma

**DOI:** 10.3390/biomedicines10071463

**Published:** 2022-06-21

**Authors:** Barbara Bonnesen, Pradeesh Sivapalan, Alexander Jordan, Johannes Wirenfeldt Pedersen, Christina Marisa Bergsøe, Josefin Eklöf, Louise Lindhardt Toennesen, Sidse Graff Jensen, Matiullah Naqibullah, Zaigham Saghir, Jens-Ulrik Stæhr Jensen

**Affiliations:** 1Section of Respiratory Medicine, Department of Medicine, Herlev and Gentofte Hospital, University of Copenhagen, 2900 Hellerup, Denmark; bbbb@dadlnet.dk (B.B.); pradeesh.sivapalan.02@regionh.dk (P.S.); alexander.svorre.jordan@regionh.dk (A.J.); christina.marisa.bergsoee@regionh.dk (C.M.B.); josefin.viktoria.ekloef@regionh.dk (J.E.); sidse.graff.jensen@regionh.dk (S.G.J.); matiullah.naqibullah@regionh.dk (M.N.); zaigham.saghir@regionh.dk (Z.S.); 2Department of Oncology, Herlev and Gentofte Hospital, University of Copenhagen, 2730 Herlev, Denmark; johannes.wirenfeldt.vad.pedersen@regionh.dk; 3Department of Respiratory Medicine, University of Copenhagen, Hvidovre Hospital, 2650 Hvidovre, Denmark; louise.toennesen@gmail.com; 4Department of Clinical Medicine, Faculty of Health Sciences, University of Copenhagen, 2200 Copenhagen, Denmark

**Keywords:** Chronic Obstructive Pulmonary Disease (COPD), asthma, asthma-COPD overlap, cancer, inhaled corticosteroids, ICS

## Abstract

Chronic inflammation such as asthma may lead to higher risks of malignancy, which may be inhibited by anti-inflammatory medicine such as inhaled corticosteroids (ICS). The aim of this study was to evaluate if patients with asthma-Chronic Obstructive Pulmonary Disease (COPD) overlap have a higher risk of malignancy than patients with COPD without asthma, and, secondarily, if inhaled corticosteroids modify such a risk in a nationwide multi-center retrospective cohort study of Danish COPD-outpatients with or without asthma. Patients with asthma-COPD overlap were propensity score matched (PSM) 1:2 to patients with COPD without asthma. The endpoint was cancer diagnosis within 2 years. Patients were stratified depending on prior malignancy within 5 years. ICS was explored as a possible risk modifier. We included 50,897 outpatients with COPD; 88% without prior malignancy and 20% with asthma. In the PSM cohorts, 26,003 patients without prior malignancy and 3331 patients with prior malignancy were analyzed. There was no association between asthma-COPD overlap and cancer with hazard ratio (HR) = 0.92, CI = 0.78–1.08, *p* = 0.31 (no prior malignancy) and HR = 1.04, CI = 0.85–1.26, and *p* = 0.74 (prior malignancy) as compared to patients with COPD without asthma. ICS did not seem to modify the risk of cancer. In conclusion, in our study, asthma-COPD overlap was not associated with an increased risk of cancer events.

## 1. Introduction

COPD is a chronic pulmonary condition often linked to current or previous smoking. It shares many risk factors with cancer development, and the two are frequently found together [1,2,3,4].

Patients with asthma seem to have an elevated level of chronic low-grade inflammation with elevated pro-inflammatory biomarkers such as high sensitivity C-reactive protein (hsCRP) [5,6], which has been shown to increase the risk of cancer [7,8,9,10]. Furthermore, various anti-inflammatory treatments have been proposed to reduce the risk of cancer.

Inhibition of Interleukin-1-beta with canakinumab has been shown to reduce the risk of cancer, especially lung cancer death, by 70% in a sub-study to a randomized controlled trial [11]. The sub-study was based on the CANTOS randomized controlled trial, which examined 10,061 patients with previous myocardial infarction and an elevated level of hsCRP, to see if inhibition of Interleukin-1-beta driven inflammation could reduce the risk of cardiovascular events [12]. Similarly, cohort studies have pointed towards a cancer-protective effect of ICS, which has also been shown to lower hsCRP in patients with asthma [13,14,15,16,17,18].

It seems plausible that asthma per se increases systemic inflammation and could increase the risk of malignancy [1,19,20,21,22,23,24]. However, only a few smaller, uncontrolled studies have examined malignancy in patients with asthma-COPD overlap [25,26]. These studies did not lead to a conclusion in risk profile among patients with asthma-COPD overlap and leave room for further investigation.

We aimed to find out whether the risk of cancer events is higher in patients with asthma-COPD overlap, as compared to those with COPD alone in a large nationwide epidemiological study, as well as to examine if ICS had a cancer-preventive effect in this population.

## 2. Materials and Methods

### 2.1. Study Design

A multi-center retrospective cohort study was conducted by combining data from the following registries:The Danish Register of Chronic Obstructive Pulmonary Disease (DrCOPD), which contains nationwide information on COPD outpatients since 2008 [27].The Danish Civil Registration System with unique personal identification of all Danish citizens [28].The Danish National Patient Registry holds information on all admissions to Danish hospitals and outpatient clinic visits [29].The Danish National Health Service Prescription Database holds information on all prescriptions dispensed in Danish pharmacies since 2004 including date of dispensation, formulation, strength, and quantity [30].

We included all Danish residents > 30 years of age (Figure 1) with COPD diagnosed by a respiratory medicine specialist in an outpatient clinic from 1 January 2010 to 24 June 2017. The study entry date was defined as the first contact to an outpatient clinic. Patients with no known tobacco exposure and patients who were diagnosed with asthma after the initial contact date were excluded. The patients were stratified into two groups depending on previous malignancy within the 5 years before inclusion (prior malignancy vs. no prior malignancy). This was done to minimize the risk of differences in outcome from differences in underlying risk of malignancy including recurrent cancer events. We chose to analyze both patients with and without prior malignancy in order to secure a full-population analysis not excluding patients with certain characteristics, while capturing all cancer events in the studied population, and hereby making sure not to miss a difference in risk profile associated to asthma due to too small a sample size.

In both the group with and the group without previous malignancy, patients with asthma-COPD overlap were propensity score-matched to two controls with COPD without asthma. The propensity score matching was performed based on the following known and likely confounders: age, gender, tobacco exposure, body mass index (BMI), medical research council (MRC), and forced expiratory volume in first second as percent of predicted (FEV_1_%) [31]. 

We registered the demographic variables, comorbidities, prior exacerbations of COPD, and medical therapy (Table 1).

Patients were followed for 2 years after study entry, during which time they were eligible to develop an event. The 2-year follow-up time was chosen to capture the developed cancer events, while still anticipating mortality as a not too dominant of a competing risk.

Cancer events were defined and registered as all diagnoses of cancer (except non-melanoma skin cancer): malignant neoplasms of lip, oral cavity and pharynx: C00–C14; digestive organs: C15–C26; respiratory and intrathoracic organs: C30–C39; bone and articular cartilage: C40–C41; malignant melanoma of skin: C43; Merkel cell carcinoma: C4A; mesothelial and soft tissue: C45–C49; breast: C50; female genital organs: C51–C58; male genital organs: C60–C63; urinary tract: C64–C68; eye, brain and other parts of central nervous system: C69–C72; thyroid and other endocrine glands: C73–C75; ill-defined, other secondary and unspecified sites: C76–C80; neuroendocrine tumors: C7A; lymphomas: C81–C88; and leukemia: C90–C96 (Table 2).

Patients without prior malignancy and without asthma were divided into four groups based on their treatment with ICS in the year prior to study entry: No ICS treatment, low dose (budesonide equivalent dose < 370 µg daily), medium dose (budesonide equivalent dose ≥ 370 µg and <970 µg daily), and high dose (budesonide equivalent dose ≥ 970 µg daily). Treatment with ICS was defined as total redeemed prescriptions of ICS recalculated to daily dose. All patients included in this analysis suffered from both asthma and COPD. However, there was a clear relation between the dosage of daily ICS use and possible confounders including older age, male sex, and lower FEV_1_% as well as exacerbations in the year prior to inclusion (Table 3). Hence, to minimize biases these patients were assigned weight based on age, gender, tobacco exposure, MRC, BMI, and FEV_1_% by inverse probability of treatment weighting before Cox analysis.

Our primary endpoint was time to cancer events as analyzed by Cox regression ana-lysis. As a few patients had more than one event during the follow-up period, only the first event was counted in Cox regression analysis and incidence plots.

We performed adjusted Cox regression analysis to evaluate the effect of comorbidities by Charlson Comorbidity Index while adjusting for gender and asthma.

We performed a sensitivity analysis by an unadjusted Cox regression analysis of the entire included population on the parameters included in the propensity score match, as well as by exacerbations within the year before inclusion.

### 2.2. Statistical Analysis

Patients with asthma-COPD overlap were propensity score matched (using the Greedy Match algorithm from the Mayo Clinic [32]) to patients with COPD without asthma by the following known and suspected confounders: age (as a continuous variable), gender, tobacco exposure (divided into the categories “passive smoking”, “previous smoking”, “active smoking” and “unknown tobacco exposure”), MRC (with the options 1, 2, 3, 4 and 5), BMI (as a continuous variable), and FEV_1_% (as a continuous variable) at inclusion.

Baseline characteristics were compared by chi-square test. Cox regression model was used to assess the risk of cancer events before and after propensity score matching.

Incidence plots were calculated with mortality due to any other cause than cancer as a competing risk, corresponding to significance by unadjusted Cox analysis and as determined by Gray’s K-sample test examining cancer events among propensity score-matched groups.

Inverse probability of treatment weighting using generalized boosted models was employed to balance covariates between the ICS treatment groups [33]. Propensity scores were calculated based on the before-mentioned confounders (age, gender, tobacco exposure, MRC, BMI, and FEV_1_%).

The relative risk of cancer events or death by any other cause was estimated using inverse probability of treatment-weighted Cox proportional hazards models. Risk of cancer events was analyzed by right-censoring competing risks. Results were presented as hazard ratios (HR) with 95% confidence intervals (CI) and cause-specific HRs with 95% CI.

An adjusted Cox proportional hazard model in the unmatched population with a follow-up period of 730 days served as sensitivity analysis. The analysis was adjusted for the variables included in the propensity score match (age, gender, tobacco exposure, MRC, BMI, and FEV_1_%).

An adjusted Cox proportional hazards analysis was performed on the group of propensity-matched patients, with adjustment for the above-mentioned parameters (age, gender, tobacco exposure, MRC, BMI, and FEV_1_%). Model control investigating the proportional hazards assumption was performed to validate the Cox proportional hazards regression on the following parameters; age, gender, tobacco exposure, MRC, BMI, and FEV_1_%. In all cases yielding *p*-values > 0.05.

All statistical analyses were performed using SAS 9.4, SAS Institute Inc, Cary, North Carolina, United States of America, and the R statistical programming language. A two-sided 95%-confidence interval was considered statistically significant. Plots were customized by the NewSurv macro [34].

### 2.3. Ethics

The study was approved by the Danish Data Protection Agency (Journal number: P-2021-280). In Denmark, retrospective use of register data does not require ethical approval or patient consent.

## 3. Results

A total of 50,897 patients were included in the study. Of these, 88.0% (44,798 patients) had no prior malignancy within 5 years before study entry. In this group, 19.7% (8828 patients) had asthma-COPD overlap compared to 18.3% (1116 patients) in the group of patients with known malignancy prior to study entry.

Baseline data of the cohorts are shown in Table 1. There were few differences in the baseline characteristics, and especially there were no clinically relevant differences in prior malignancy. However, the prevalence of atopy was markedly higher among patients with asthma-COPD-overlap (no prior malignancy: 15.2% vs. 3.2%, *p* < 0.0001 and with prior malignancy: 14.0% vs. 3.7%, *p* < 0.0001), and there was a slightly higher prevalence of patients with asthma-COPD overlap diagnosed with osteoporosis/osteopenia. Additionally, patients with asthma-COPD overlap were to a clinically relevant degree more likely to have had an admission due to an exacerbation within the year before inclusion, and of having been treated with oral corticosteroids, ICS, and long- and short-acting β2-agonists (LABA and SABA) within the year before inclusion than patients with COPD without asthma.

The risk of a cancer event was not increased in patients with asthma-COPD overlap compared to propensity score-matched patients with COPD without asthma by Cox analysis in patients without prior malignancy (HR 0.92 CI 0.78–1.08, *p* = 0.31) and neither in patients with prior malignancy (HR 1.04, CI 0.85–1.26, *p* = 0.74) (Table 2).

The risk of a lung cancer event was also not increased in patients with asthma-COPD overlap (no prior malignancy: HR 0.77, CI 0.56–1.07, *p* = 0.12; and prior malignancy: HR 0.68, CI 0.42–1.10, *p* = 0.11). In addition, there was no clinically relevant difference dependent on concomitant asthma for any other subtypes of cancer shown in Table 2 (HR, CI, and *p*-values not shown).

Two-year mortality from non-malignancy causes was not statistically different: HR 1.04 in the groups without prior malignancy (1347 succumbed within two years with asthma-COPD overlap and 2786 patients with COPD without asthma, *p* = 0.31), and HR 0.89 in the groups with prior malignancy (162 succumbed with asthma-COPD overlap and 339 patients without asthma, *p* = 0.21).

Cumulative incidence plots similarly showed no difference in risk of cancer events (Figure 2) (no prior malignancy HR 1.01, CI 0.86–1.19, *p* = 0.91 and prior malignancy HR 0.93, CI 0.76–1.13, *p* = 0.49) by Grays test.

The adjusted Cox regression analysis evaluating the effect of comorbidities showed no interaction between Charlson Comorbidity Index score and risk of all cancer events (HR 0.97, CI 0.87–1.00, *p* = 0.65), and the sensitivity analysis evaluating the effect of exacerbations in the last year prior to inclusion also showed no interaction with risk of all cancer events (HR 0.96, CI 0.87–1.06, *p* = 0.40).

In total, 35,970 patients without asthma and prior malignancy (70.7% of the included population) were available for analysis of ICS dosage association to cancer. Their baseline data are shown in Table 3.

Patients treated with low- or medium-dose ICS in the year before study entry had a similar risk of cancer events to patients with no ICS treatment. Patients on high-dose ICS had a borderline lowered risk of all cancer events (HR 0.96, CI 0.93–0.99, *p* = 0.01) and a slightly higher mortality from non-malignancy causes (HR 1.03, CI 1.02–1.04, *p* < 0.0001) (Table 4). There was no difference in the risk of lung cancer in the different ICS treatment groups.

Loss to follow-up was seldom; *N* = 66 (0.2%) were lost to follow-up and no meaningful differences were observed among groups.

The sensitivity analysis of the entire included population on the parameters included in the propensity score match (age, gender, tobacco exposure, MRC, BMI, and FEV_1_%) yielded the following results by Cox multivariate analysis: Age: HR 1.00, CI 1.00–1.01, and *p* = 0.83; gender: HR 0.96, CI 0.90–1.01 and *p* = 0.14; tobacco exposure: HR 1.12, CI 1.08–1.17, and *p* < 0.0001; MRC: HR 1.08, CI 1.05–1.11, and *p* < 0.0001; BMI: HR 0.99, CI 0.98–0.99, and *p* < 0.0001; and FEV_1_%: HR 1.00, CI 1.00–1.00, and *p* = 0.64.

## 4. Discussion

In this nationwide register-based cohort study, outpatients with asthma-COPD overlap did not have an increased risk of cancer, including lung cancer events when compared to patients with COPD without asthma.

Our results were consistent across analyses, study cohorts, and outcomes. We analyzed patients without and with previous cancer within 5 years before study entry, and even across these analyses, there was no signal of an increased or decreased risk of cancer among patients with asthma-COPD overlap compared to patients with COPD without asthma.

Inverse propensity-weighted treatment group analysis did not reveal any clinically significant association between ICS administration and development of cancer among patients with COPD without asthma.

To our knowledge, our study was the largest study to investigate the risk of cancer events among patients with asthma-COPD overlap. Additionally, the certainty of the diagnoses was high in our study: COPD and asthma diagnoses were verified by specialist and spirometry. Finally, we had complete follow-up and the possibility to control for important confounders like smoking status, spirometry measures, BMI, age, and others.

We also had available to us all diagnoses set by a hospital physician within co-morbidities, and though it is not possible to propensity score match or adjust for all baseline characteristics, we could account for all serious co-morbidities at baseline.

In Danish registers of all hospitalizations, all visits are to out-patient clinics and all prescription medicine are purchased in pharmacies, and hence we can assume that the number of patients who developed cancer during the follow-up period without a diagnosis registered in our study is very few.

Previous studies in patients with asthma-COPD overlap include a large Swedish cohort study from primary care with 19,894 patients with COPD of whom 3597 also suffered from asthma. In contrast to our study, this study found that patients who developed pulmonary cancer had a lower prevalence of asthma before the time of COPD diagnosis. However, a weakness of this study was the lack of specialist verified respiratory diseases diagnoses [25]. Another study examining associations between asthma-COPD overlap and cancer is a Canadian cross-sectional study, which found that cancer was more prevalent in patients with asthma-COPD overlap than in patients with COPD without asthma. This study naturally lacks the element of time, it was not able to adjust for confounders, and the groups of patients with asthma-COPD overlap and with COPD without asthma were in many ways not comparable [26].

Taken together, neither ICS use, nor specifically the dose used, was associated with any clinically meaningful risk changes for cancer, although some risk estimates had low *p*-values. Lowering chronic inflammation may decrease the risk of cancer [11,13,14,15,16,17,18], and thus the more frequent use of oral and inhaled corticosteroids in patients with asthma-COPD overlap may have decreased their risk of cancer events by lowering chronic inflammation. Since our results were neutral, taken together, this could be caused, at least in part, by “neutralization” of a possible increased risk of cancer events in the asthma-COPD overlap patients on one side, and a protective effect of oral corticosteroids and ICS in the same groups of patients on the other side.

Despite some strengths, our study has some limitations that deserve careful evaluation. Biases and a possible neutralization effect cannot be disentangled completely, because of the observational nature of the design. Additionally, our knowledge on tobacco exposure could not be quantified as pack years and was limited to “Passive smoking”, “Previous smoker”, “Active smoker”, and “Unknown tobacco exposure”, although the latter did only comprise one tenth of the patients. Adjusting for tobacco exposure as a continuous variable quantified as packyears would have been desirable; however, with our available databases, this was not possible. A study looking at reliability and validity of tobacco exposure history has shown a similar, high level of agreement (Kappa values 0.92–1.00) for use of cigarettes quantified similar to our method and as lifetime use. In contrast to cigarettes, use of other types of tobacco was less consistently reported both as current use and lifetime use at the different timepoint [35]. Additionally, a study has shown that semi-quantifying tobacco exposure in patients with asthma-COPD overlap is sufficient for predicting emphysema [36].

We did not add all baseline variables as matching variables in the matched cohort and as adjusting variables in the Cox analyses, since not all these variables were judged to have confounding potential, and since adding too many may have weakened the power of the models. Further, we could have overfitted the model. We conducted subgroup analysis of all endpoints, which may have caused us to miss minor differences depending on asthma status among less-frequent secondary endpoints in the small subgroups, such as the risk of pulmonary cancer among patients with prior cancer.

In conclusion, asthma in patients with COPD does not seem to be associated with an increased risk of cancer events. ICS did not, to a clinically relevant degree, influence the risk of cancer events in patients with COPD-patients asthma. Hence, our results suggest that neither a concomitant diagnosis of asthma nor use of ICS should be considered independent risk or protective factors when assessing a patient with COPD and possible cancer. As a consequence, our data do not support a different approach for cancer diagnostic workup or screening among patients with asthma-COPD-overlap as compared to patients with COPD without asthma.

## Figures and Tables

**Figure 1 biomedicines-10-01463-f001:**
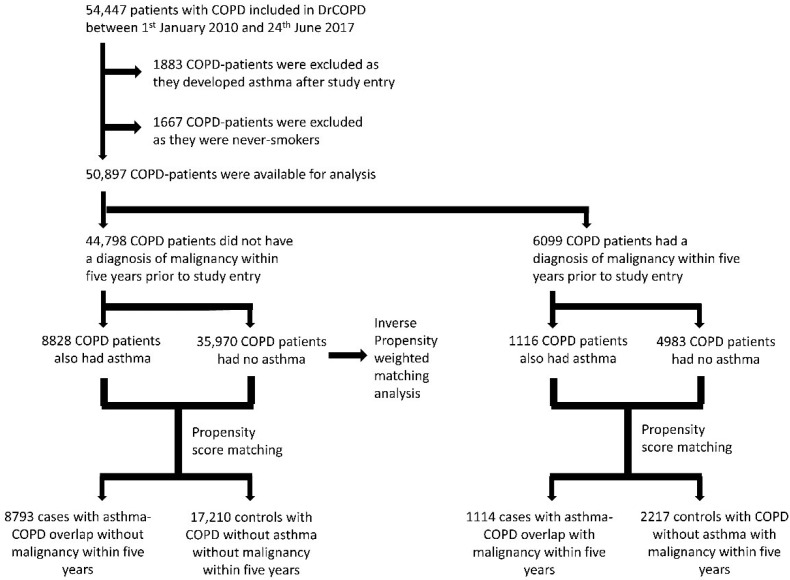
Flowchart of included patients.

**Figure 2 biomedicines-10-01463-f002:**
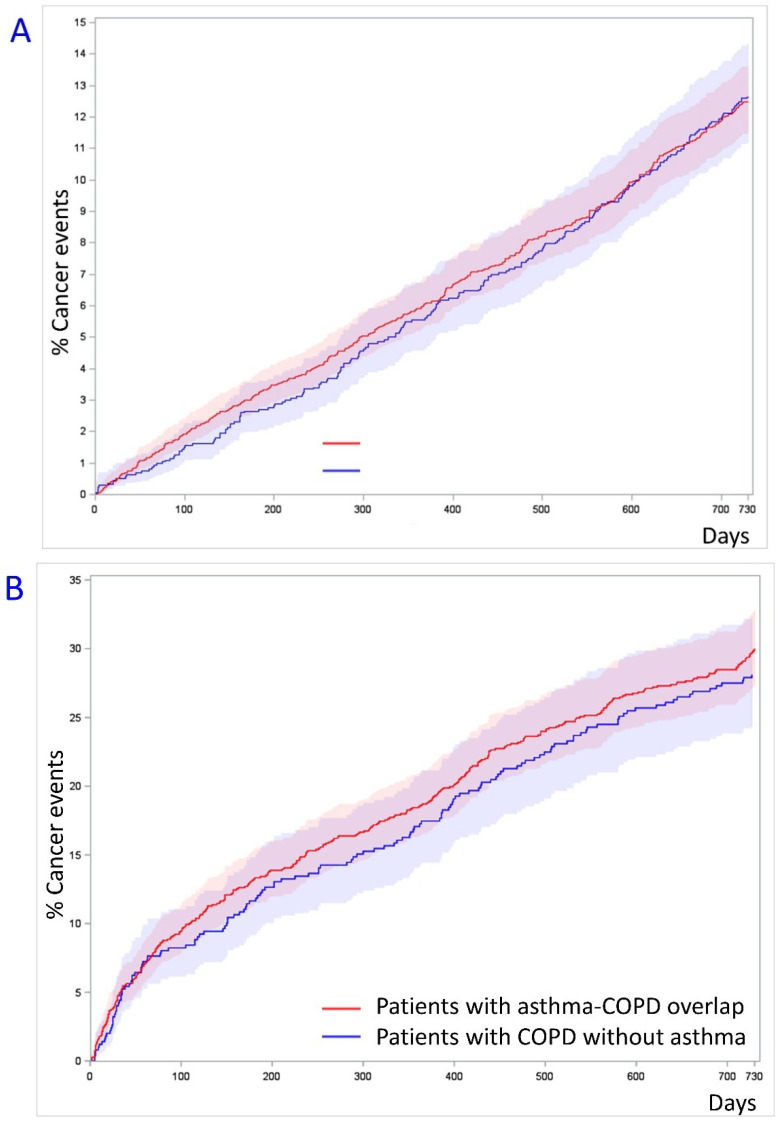
Cumulated incidence plots of cancer events occuring to propensity-matched cohorts. (**A**): Cancer events in the group of patients without prior malignancy within 5 years (*N* = 26,003). (**B**): Cancer events in the group of patients with prior malignancy within 5 years (*N* = 3331).

**Table 1 biomedicines-10-01463-t001:** Characteristics of propensity-matched groups of patients with asthma-COPD overlap and groups of patients with COPD without asthma.

Characteristic	No Previous Cancer *		Previous Cancer *	
	Patients with Asthma-COPD Overlap (*N* = 8793)	Patients with COPD without Asthma (*N* = 17,210)	Patients with Asthma-COPD Overlap (*N* = 1114)	Patients with COPD without Asthma (*N* = 2217)
Age, years	67.7 (58.7–76.0)	68.0 (59.9–75.6)	71.9 (65.4–77.9)	71.9 (65.5–78.3)
Gender, female	5074 (57.7)	9746 (56.6)	650 (58.3)	1287 (58.1)
Tobacco exposure:				
Passive smoking	1 (0.0)	0 (0.0)	1 (0.1)	0 (0.0)
Previous smoker	5157 (58.6)	9745 (56.6)	678 (60.9)	1301 (58.9)
Active smoker	2881 (32.8)	6186 (35.9)	308 (27.6)	682 (30.8)
Unknown tobacco exposure	754 (8.6)	1279 (7.4)	127 (11.4)	234 (10.6)
MRC	3 (2–4)	3 (2–4)	3 (3–4)	3 (3–4)
BMI	25 (22–29)	25 (22–29)	25 (22–28)	25 (22–28)
FEV1 in percentage of expected	49 (36–61)	49 (36–61)	49 (36–59)	49 (36–59)
Malignancy within five years:			1114 (100.0)	2217 (100.0)
Oropharyngeal cancer			82 (7.4)	135 (6.1)
Intestinal cancer			139 (12.5)	256 (11.5)
Pancreas cancer			9 (0.8)	9 (0.4)
Other digestive organ cancer			41 (3.7)	53 (2.4)
Lung cancer			221 (19.8)	537 (24.2)
Other respiratory and intrathoracic cancers			50 (4.5)	82 (3.7)
Malignant melanomas			29 (2.6)	49 (2.2)
Mesothelial, soft tissue, bone, cartilage, nervous system, eye and brain cancers			29 (2.6)	42 (1.9)
Malignant neoplasms of the breast			190 (17.1)	422 (19.0)
Female genital organ cancers			54 (4.8)	81 (3.7)
Male genital organ cancers			148 (13.3)	298 (13.4)
Kidney and urinary tract cancers			84 (7.5)	169 (7.6)
Endo- and neuroendocrine cancers			7 (0.6)	18 (0.8)
Other cancers (non-defined)			122 (11.0)	279 (12.6)
Malignant lymphomas			45 (4.0)	123 (5.5)
Malignant leukemia			63 (5.7)	130 (5.9)
Concomitant cancers			199 (17.9)	447 (20.2)
Comorbidities:				
Hypertension	2747 (31.2)	5105 (29.7)	452 (40.6)	838 (37.8)
Hypercholesterolemia	1075 (12.2)	2152 (12.5)	201 (18.0)	335 (15.1)
Atrial fibrillation	1155 (13.1)	2294 (13.3)	210 (18.9)	415 (18.7)
Diabetes	1135 (12.9)	1990 (11.6)	165 (14.8)	287 (12.9)
Osteoporosis or osteopenia	1956 (22.2)	3104 (18.0)	341 (30.6)	568 (25.6)
Renal insufficiency	3086 (35.1)	5710 (33.2)	500 (44.9)	908 (41.0)
Liver insufficiency	264 (3.0)	528 (3.1)	41 (3.7)	85 (3.8)
Atopy or allergy	1338 (15.2)	552 (3.2)	156 (14.0)	81 (3.7)
Depression	512 (5.8)	760 (4.4)	78 (7.0)	114 (5.1)
Exacerbations requiring admission within the last year prior to inclusion	3017 (34.3)	5005 (29.1)	428 (38.4)	689 (31.1)
Medical treatment for respiratory disease within the last year prior to inclusion:				
Oral corticosteroid	4932 (56.1)	6958 (40.4)	670 (60.1)	970 (43.8)
Inhaled corticosteroid (ICS)	7744 (88.1)	11,647 (67.8)	994 (89.2)	1508 (68.0)
Long acting β2-agonist (LABA)	7679 (87.3)	12,743 (74.0)	1005 (90.2)	1690 (76.2)
Long acting muscarinic receptor antagonist (LAMA)	6246 (71.0)	11,954 (69.5)	861 (77.3)	1595 (71.9)
Short acting β2-agonist (SABA)	6774 (7.0)	10,886 (63.3)	892 (80.1)	1389 (62.7)
Short acting muscarinic receptor antagonist (SAMA)	371 (4.2)	515 (3.0)	68 (6.1)	100 (4.5)

Characteristics are presented as medians and absolute numbers as relevant with interquartile ranges and percentages in parenthesis. Patients with asthma-COPD overlap were propensity score matched 1:2 to groups of patients with COPD without asthma by the following parameters: age, gender, tobacco exposure, MRC, BMI, and FEV_1_%. * Previous cancer was defined as active cancer within 5 years before inclusion.

**Table 2 biomedicines-10-01463-t002:** Cancer events in propensity-matched groups of patients with asthma-COPD overlap and groups of patients with COPD without asthma.

Endpoint	Patients without a Diagnosis of Malignancy within 5 Years Prior to Inclusion		Patients with a Diagnosis of Malignancy within 5 Years Prior to Inclusion	
	Patients with Asthma-COPD Overlap (*N* = 8793)	Patients with COPD without Asthma (*N* = 17,210)	Patients with Asthma-COPD Overlap (*N* = 1114)	Patients with COPD without Asthma (*N* = 2217)
Cancer events				
*N* (%)	219 (2.5)	457 (2.7)	140 (12.6)	335 (15.1)
^#^ HR	0.92 (0.78–1.08)	Reference	1.04 (0.85–1.26)	Reference
Lung cancer events				
*N* (%)	52 (0.6)	129 (0.7)	21 (1.9)	77 (3.5)
^#^ HR	0.77 (0.56–1.07)	Reference	0.68 (0.42–1.10)	Reference
Cancer events:				
Oropharyngeal cancer	4 (0.0)	13 (0.1)	1 (0.1)	5 (0.2)
Intestinal cancer	35 (0.4)	63 (0.4)	10 (0.9)	19 (0.9)
Pancreas cancer	3 (0.0)	2 (0.0)	0 (0.0)	2 (0.1)
Other digestive organ cancers	9 (0.1)	29 (0.2)	2 (0.2)	9 (0.4)
Other respiratory and intrathoracic cancers	2 (0.0)	12 (0.1)	1 (0.1)	2 (0.1)
Mesothelial, soft tissue, bone, cartilage, nervous system, eye, and brain cancers	1 (0.0)	7 (0.0)	3 (0.3)	4 (0.2)
Malignant melanomas	6 (0.1)	5 (0.0)	1 (0.1)	5 (0.2)
Malignant neoplasms of the breast	40 (0.5)	72 (0.4)	28 (2.5)	47 (2.1)
Female genital organ cancers	4 (0.0)	6 (0.0)	2 (0.2)	2 (0.1)
Male genital organ cancers	21 (0.2)	52 (0.3)	26 (2.3)	59 (2.7)
Urinary tract cancers	21 (0.2)	24 (0.1)	12 (1.1)	21 (0.9)
Endo- and neuroendocrine cancers	3 (0.0)	3 (0.0)	2 (0.2)	2 (0.1)
Other cancers (non-defined)	6 (0.1)	5 (0.0)	11 (1.0)	24 (1.1)
Malignant lymphomas	6 (0.1)	17 (0.1)	6 (0.5)	28 (1.3)
Malignant leukemia	6 (0.1)	18 (0.1)	14 (1.3)	29 (1.3)
Non-malignancy related mortality				
*N* (%)	1347 (15.3)	2786 (16.2)	162 (14.5)	339 (15.3)
^#^ HR	1.04 (0.97–1.10)	Reference	0.89 (0.73–1.07)	Reference

Results are presented as absolute numbers and ratios as relevant with percentages and 95% confidence intervals in parenthesis. ^#^ as analyzed by unadjusted Cox regression analysis.

**Table 3 biomedicines-10-01463-t003:** Characteristics of ICS treatment groups of patients with COPD without asthma and without previous malignancy within 5 years prior to inclusion.

Characteristic	No Inhaled Corticosteroid (*N* = 11,665)	Low-Dose Inhaled Corticosteroid (*N* = 9309)	Medium-Dose Inhaled Corticosteroid (*N* = 6761)	High-Dose Inhaled Corticosteroid (*N* = 8235)	*p*-Value for Difference between ICS Treatment Groups
Age, years	69.5 (61.2–77.2)	71.0 (62.9–78.4)	72.5 (65.4–79.1)	72.7 (65.8–79.2)	<0.0001 * <0.0001 ** <0.0001 ***
Gender, female	5379 (46.1)	4593 (49.3)	3522 (52.1)	4731 (57.4)	<0.0001
Tobacco exposure:					<0.0001
Passive smoking	0 (0.0)	0 (0.0)	0 (0.0)	0 (0.0)	
Previous smoker	5463 (46.1)	5171 (55.5)	4300 (63.6)	5250 (63.8)	
Active smoker	4628 (39.7)	3384 (36.4)	2057 (30.4)	2552 (31.0)	
Unknown tobacco exposure	1574 (13.5)	754 (8.1)	404 (6.0)	433 (5.3)	
MRC	3 (2–3)	3 (2–3)	3 (3–4)	3 (3–4)	<0.0001 * <0.0001 ** <0.0001 ***
BMI	25 (22–28)	25 (22–29)	25 (21–28)	25 (21–27)	=0.41 * <0.0001 ** <0.0001 ***
FEV1 in percentage of expected	54 (47–69)	49 (40–62)	44 (33–55)	39 (29–49)	<0.0001 * <0.0001 ** <0.0001 ***
Comorbidities:					
Hypertension	3824 (32.8)	3046 (32.7)	2146 (31.7)	2517 (30.6)	0.003
Hypercholesterolemia	1837 (15.7)	1352 (14.5)	789 (11.7)	830 (10.1)	<0.0001
Atrial fibrillation	1821 (15.6)	1563 (16.8)	1040 (15.4)	1149 (14.0)	<0.0001
Diabetes	1473 (12.6)	1142 (12.3)	702 (10.4)	831 (10.1)	<0.0001
Osteoporosis or osteopenia	1662 (14.2)	1476 (15.9)	1393 (20.6)	2094 (25.4)	<0.0001
Renal insufficiency	4227 (36.2)	3347 (36.0)	2310 (34.2)	2748 (33.4)	<0.0001
Liver insufficiency	385 (3.3)	276 (3.0)	167 (3.3)	175 (2.1)	<0.0001
Atopy or allergy	354 (3.0)	289 (3.1)	184 (2.7)	194 (2.4)	0.01
Depression	496 (4.3)	442 (4.7)	313 (4.6)	371 (4.5)	0.35
Exacerbations requiring admission within the last year prior to inclusion	1939 (16.6)	2932 (31.5)	2473 (36.6)	3480 (42.3)	<0.0001
Medical treatment for respiratory disease within the last year prior to inclusion:					
Oral corticosteroid	2444 (21.0)	3697 (39.7)	3546 (52.4)	5107 (62.0)	<0.0001
Long acting β2-agonist (LABA)	3317 (28.4)	8686 (93.3)	6580 (07.3)	8079 (98.1)	<0.0001
Long acting muscarinic receptor antagonist (LAMA)	5635 (48.3)	6531 (70.2)	5711 (84.5)	7316 (88.8)	<0.0001
Short acting β2-agonist (SABA)	4304 (36.9)	6118 (65.7)	5163 (76.4)	6910 (83.9)	<0.0001
Short acting muscarinic receptor antagonist (SAMA)	215 (1.8)	260 (2.8)	289 (4.3)	339 (4.1)	<0.0001

Characteristics are presented as medians and absolute numbers as relevant with interquartile ranges and percentages in parenthesis. *p*-values were calculated by Chi2 test or general linear model. * Low dose ICS vs. no ICS, ** Medium-dose ICS vs. no ICS, *** High-dose ICS vs. no ICS.

**Table 4 biomedicines-10-01463-t004:** Cancer events of patients with COPD without asthma and without previous malignancy within 5 years prior to inclusion.

Endpoint	No Inhaled Corticosteroid (*N* = 11,665)	Low-Dose Inhaled Corticosteroid (*N* = 9309)	Medium-Dose Inhaled Corticosteroid (*N* = 6761)	High-Dose Inhaled Corticosteroid (*N* = 8235)
Cancer events	Reference			
^#^ HR		1.01 (0.92–1.11)	0.97 (0.92–1.01)	0.96 (0.93–0.99) *
*p*-value		0.88	0.12	0.01
Lung cancer events	Reference			
^#^ HR		1.09 (0.90–1.31)	0.96 (0.89–1.05)	1.00 (0.94–1.05)
*p*-value		0.37	0.41	0.86
Non-malignancy related mortality	Reference			
^#^ HR		1.04 (1.01–1.09) *	1.01 (1.00–1.03)	1.03 (1.02–1.04)
*p*-value		0.03	0.15	* *p* < 0.0001

Results are presented as absolute numbers and ratios as relevant with percentages and 95% confidence intervals in parenthesis. ^#^ as analyzed by unadjusted Cox regression analysis. * indicates statistical significance > 0.95 by regression analysis.

## Data Availability

Data is available in the following registers. (1) The Danish Register of Chronic Obstructive Pulmonary Disease (DrCOPD), which contains nationwide information on COPD outpatients since 2008 [27]. (2) The Danish Civil Registration System with unique personal identification of all Danish citizens [28]. (3) The Danish National Patient Registry holds information on all admissions to Danish hospitals and outpatient clinic visits [29]. (4) The Danish National Health Service Prescription Database holds information on all prescriptions dispensed in Danish pharmacies since 2004 including date of dispensation, formulation, strength, and quantity [30].

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
