# Peer review of "Risk of Malignancy in Patients with Asthma-COPD Overlap Compared to Patients with COPD without Asthma"

_biomedicines, 2022, doi:10.3390/biomedicines10071463_

Round 1
Reviewer 1 Report
Dear Authors,
I have read the manuscript and I send you my comments:
1) Please evaluate the data considering a subanalysis respect to age, comorbidity and gender
2) Discussion is very short, please rewrite
Author Response
Thank you for your comments. Please see attached.

Reviewer 2 Report
The retrospective use of this large data set is justified in answering the questions posed by the authors. The paper is well written, the data analysed and presented appropriately. The conclusions are supported by the data and discussed clearly in the paper.
Line 164: above mentioned.
Author Response

(The authors gave the same response as above.)

Reviewer 3 Report
1. In patient with COPD, cancer risk from smoking exposure is much higher that for systemic inflammation caused asthma. However, in multivariate regression analysis, the consideration for smoking exposure is insufficient.
2. In table 3, please show the p-value of each variable.
3. Is there the specific reason for including the patients having the prior malignancy?
4. In COPD, there are insufficent references on the association between the chronic inflammatory disease such as asthma and other cancers, and you could not fully describe the adequate explanations about the association. no adequate exp

Author Response

(The authors gave the same response as above.)

Round 2
Reviewer 1 Report
Dear Authors,
I have read the manuscript.
I think that it has been revised, and can be published.
There are some typo errors that can be revised during the editing
Reviewer 3 Report
Thank you for the faithful revision. I do not have further comments about your manuscript.